# Inflammatory Joint Pathologies and the Oral–Gut Microbiota: A Reason for Origin

**DOI:** 10.3390/healthcare13161942

**Published:** 2025-08-08

**Authors:** Mario Salazar-Páramo, Fabiola de Santos Ávila, Genaro E. Ortiz-Velázquez, Ian Ramirez-Jaramillo, Daniela L. C. Delgado-Lara, Erandis Dheni Torres-Sánchez, Genaro Gabriel Ortiz

**Affiliations:** 1Department of Physiology, Health Science Center University, University of Guadalajara, Guadalajara 44100, Jalisco, Mexico; mario.sparamo@academicos.udg.mx; 2Department of Public Health, Health Science Center University, University of Guadalajara, Guadalajara 44100, Jalisco, Mexico; fabiola.desantos@academicos.udg.mx; 3Emergency Clinical Laboratory, General Hospital of Zone 46 Mexican Social Security Institute (IMSS), Guadalajara 44100, Jalisco, Mexico; genaroeov@hotmail.com; 4Neurosciences Division, Western Biomedical Research Center, Mexican Social Security Institute (IMSS), Guadalajara 44100, Jalisco, Mexico; ramirezian@hotmail.com; 5Departamento Académico de Formación Universitaria, Ciencias de la Salud, Universidad Autónoma de Guadalajara, Zapopan 45129, Jalisco, Mexico; daniela.delgadolara@gmail.com; 6Department of Medical and Life Sciences, University Center of la Cienega, University of Guadalajara, Ocotlan 47820, Jalisco, Mexico; 7Department of Philosophical and Methodological Disciplines, Health Science Center University, University of Guadalajara, Guadalajara 44100, Jalisco, Mexico

**Keywords:** gut microbiome, immune response, inflammation, rheumatoid arthritis, leaky gut

## Abstract

The human gut microbiota, which can weigh as much as 2 kg and harbor 100 trillion bacteria, is specific to each individual. In healthy adults, a balanced microbiota—a state known as eubiosis—can be altered by various factors such as diet and lifestyle. Microbiota imbalance—or dysbiosis—can have consequences for host health. Given that 80% of the human immune system is located in the gut, studies have investigated the role of the microbiota in immune system diseases, including joint and inflammatory pathologies such as rheumatoid arthritis. A better understanding of this pathology might enable the development of new treatments in the future. The microbiota includes all unicellular organisms in the digestive tract, including bacteria, viruses, fungi, and archaea. This complex ecosystem is unique to each individual. Associations between the human body and the microorganisms that it hosts can be considered mutualistic, symbiotic, or parasitic. These microorganisms are responsible for essential functions in maintaining health; the microbiota can even be considered another organ of the body. Microbiota composition varies considerably between early life and older age but remains relatively stable for most of a lifespan.

## 1. Introduction

The intestinal microbiota is composed of approximately 1 × 10^14^ microorganisms per individual and may include as many as 160 of the approximately 1000 bacterial species existing in humans. These bacteria comprise four groups: Actinobacteria, Proteobacteria, Firmicutes, and Bacteroides. The latter two are the most important, representing 60–75% and 30–40%, respectively [1]. Researchers previously believed that each human had a defined pool of bacteria and that these were identical among individuals. Stool studies often revealed the presence of *E. coli*, which is currently known to represent less than 1% of all microorganisms in human intestines [1]. The Bacteroides family is among the best represented and best known. These bacteria enable the digestion of carbohydrates and other foods by producing enzymes necessary for nutrient assimilation. Bacteroides are notably present in people who consume meat and sausages [1]. In contrast, the *Prevotella* family is found primarily in people with a vegetarian diet, as well as in those who consume relatively little meat. These bacteria are often accompanied by the presence of *Desulfovibrionales* bacteria. The *Prevotella* family produces sulfur compounds, whereas *Desulfovibrionales* eliminate sulfur compounds. The *Ruminococcus* family is involved in the production of heme, a structure containing an iron atom and a constituent of hemoglobin, which is vital in transporting oxygen in the blood [2].

Although the entire human organism is colonized, the oral microbiota has a particularly important role at the gateway to the digestive system. The oral cavity is home to one of the human body’s richest microbial communities, including bacteria, viruses, yeasts, and protozoa [3]. The mouth is a complex anatomical environment that contains a variety of structures and tissues, including the teeth, gingival canal, sulcus, and mucous palate. Each of these structures represents an ecological niche with specific growth conditions and nutrients and its own microbial community. Despite showing overall similarities, microbial populations markedly differ according to their location within the oral cavity. To thrive here, the microbiota requires surface nutrients from the host. Therefore, it organizes itself into ecological niches at the mucosal or dental surface level via a biofilm architecture. Consequently, the oral microbiota comprises a set of diverse microbial biofilms [4]. The organization of microorganisms within these biofilms comprises aerobic and anaerobic bacteria adhered to oral surfaces and embedded in a microcellular salivary matrix. The biofilms further develop on non-desquamative surfaces, such as dental tissue. At the biofilm core, bacteria deploy various strategies:(a)Physical interactions that allow bacteria to adhere to each other, regardless of whether they belong to the same species, thus ensuring cohesion.(b)Nutrient exchange, wherein some bacteria release metabolites that are used by other bacteria, thus forming an interbacterial food chain.(c)Chemical communications that ensure regulation and enable coordinated behaviors within the biofilm.(d)Gene transfers that increase biofilm complexity.

Biofilm bacteria can resist the host immune response and are much more resistant to antibiotics than planktonic bacteria [5]. In healthy humans with good oral hygiene, microbiota composition remains stable. However, its balance remains fragile and depends on various factors, such as sugar and acid intake and tooth brushing [5].

The microbiota prevents the adhesion of pathogenic microorganisms through competition and notably stimulates the local immune system. The commensal microbiota therefore plays an important role in maintaining oral and systemic health. Indeed, certain microorganisms in the oral cavity can cause a range of oral pathologies such as caries, chronic periodontitis (CP), endodontic infections, alveolar osteitis, and even tonsillitis. Furthermore, several studies have linked oral pathologies to chronic systemic pathologies such as cardiovascular disease, stroke, premature birth, diabetes, and pneumonia [4]. The oral microbiota communicates with other microbiota, particularly digestive and respiratory, given the anatomical proximity of these two systems. This communication facilitates the translocation of pathogenic microorganisms. In healthy individuals with good dietary and oral hygiene habits, the oral microbiota lives in symbiosis with the host. The oral microbiota prevents the colonization of foreign pathogens and contributes to host physiology. This symbiosis is characterized by specific microorganism composition, activity, and ecology, thereby maintaining a balanced relationship with the host while being controlled by a tolerogenic immune system. However, alterations in the oral, intestinal, or other microbiota caused by certain endogenous and/or exogenous factors can lead to a shift in composition toward a dysbiotic state [4,5].

Recent evidence suggests that alterations in the oral and gut microbiota may contribute to the initiation and progression of inflammatory joint diseases, especially rheumatoid arthritis (RA). Dysbiosis has been linked to increased intestinal permeability and systemic immune activation [6]. For instance, microbial species such as *Prevotella copri*, *Collinsella aerofaciens*, and *Eggerthella lenta* have been associated with inflammatory profiles, enhanced Th17 cell differentiation, and increased protein citrullination. These processes are all involved in the autoimmune cascade observed in RA. Additionally, molecular mimicry between bacterial peptides and self-antigens may lead to the activation of autoreactive T cells, further fueling inflammation [7]. Given the bidirectional cross-talk between the gut and joint, as well as the oral–gut axis, studying these microbiota-related mechanisms could provide new insights for therapeutic strategies aiming to modulate inflammation and disease activity.

The purpose of this review is to synthesize current evidence on the role of gut microbiota and RA pathogenesis, elucidate possible mechanistic links, and discuss future directions for research and clinical applications in an autoimmune disease context. A review of this emerging relationship is essential in order to clarify its mechanisms, evaluate therapeutic prospects, and guide future research in immunopathogenesis and personalized care involving interdisciplinary collaboration between rheumatology, microbiology, and immunology.

This narrative review was developed using a structured methodology to ensure an adequate, rigorous, and well-founded selection of studies. An exhaustive search of the PubMed, Scopus and Google Scholar databases was carried out on publications from the last 20 years. We searched from 1 January 2025, to 29 April 2025. We included publications in English and Spanish that met the following criteria: observational, experimental, and clinical designs, as well as other reviews. The following terms were considered for the search: “rheumatoid arthritis”, “gut microbiome”, “immune response and gut microbiome”, “dysbiosis and inflammatory diseases”, “oral dysbiosis and inflammatory diseases”, “chronic periodontitis and rheumatoid arthritis”.

## 2. Gut Microbiota and the Immune System

The gut microbiota develops in the first 2 or 3 years of a child’s life as the immune system matures, and both processes are closely related. The intestinal immune response is localized in the mucosa, which is composed of the epithelium, the lamina propria, and the muscularis mucosae. Two distinct immune compartments are formed by intraepithelial lymphocytes and those of the lamina propria [8]; epithelial cells also participate in the body’s defense by secreting microbial peptides from Paneth cells, secreting mucus from goblet cells, and forming tight junctions that keep the intestinal barrier intact [9,10]. Figure 1 visually illustrates the structure of the intestinal mucosal barrier and the interaction between microbiota, epithelial cells, and immune components. From the lumen to the muscularis mucosae, there are key elements such as the outer and inner mucus layers, which contain microbial metabolites like short-chain fatty acids (SCFAs), neuroactive molecules, and lipopolysaccharide (LPS). These compounds play an important role in signaling between microbes and the host immune system. The inner mucus layer includes antimicrobial proteins secreted by Paneth cells and mucin glycoproteins produced by goblet cells, which help prevent bacterial translocation. The epithelial lining is shown with tight junctions, acting as a physical barrier to restrict the passage of microorganisms. In the lamina propria, immune cells such as macrophages, neutrophils, and T lymphocytes respond to potential threats. The figure also represents the process by which dendritic cells can sample antigens from the lumen and migrate to mesenteric lymph nodes to activate IgA-producing B cells, contributing to immune tolerance and regulation of the microbial population.

### Intestinal Immune Response

At the mucosa and submucosa, the intestine comprises gut-associated lymphoid tissue (GALT) or associated lymphoid structures covered with an epithelium composed of M cells that enable the capture and subsequent transport of antigens to underlying immune cells, including dendritic cells [10,11]. Peyer patches are also involved in the intestinal immune system in the small intestine, and are composed of numerous foci of B lymphocytes and a somewhat smaller number of T lymphocytes. Solitary isolated lymphoid tissue (SILT) structures are found throughout the intestine, showing the highest density in the distal colon. The link between the microbiota and immunity has been studied primarily in axenic mice; several immune system abnormalities have been found when compared with mice with a functional microbiota [10,11]. Figure 2 represents the intestinal immune structures and their interaction with the microbiota. It shows how enterocytes, goblet cells, and Paneth cells form the epithelial layer, while immune cells located in structures such as Peyer’s patches and SILT participate in the recognition and processing of microbial antigens. M cells facilitate the uptake of bacteria and their delivery to dendritic cells, which then migrate to mesenteric lymph nodes to initiate adaptive immune responses. Dendritic cells present antigens to naïve T cells and promote their differentiation into various subsets—including Th1, Th2, Th17, and Treg cells—depending on the cytokine environment. These activated T cells can then enter systemic circulation and influence immune responses beyond the gut, highlighting the relevance of the gut–immune axis in inflammatory processes.

## 3. Influence of the Microbiota on Joint Pathologies

Axenic mice are devoid of microbiota and are raised in isolation in sterile enclosures. Studies in these animals have demonstrated the influence of the gut microbiota on the host. Axenic mice have immature immune systems; they show Peyer patch hypoplasia, as well as diminished intraepithelial lymphocyte numbers, cytokine production, and serum immunoglobulin concentrations. However, these abnormalities are corrected within several weeks after inoculation with microbiota from healthy mice [12,13]. More recent studies have analyzed the relationship between the gut microbiota and RA. The microbiota depends on its host, just as the host depends on the specific and adapted microbiota. The components of a healthy microbiota are in symbiosis with one another [13,14,15].

Dysbiosis, or abnormal microbiota composition, comprises three types:(a)An absence of beneficial microorganisms;(b)An excess of pathobiont microorganisms, i.e., bacteria with roles in both eubiosis and dysbiosis;(c)A loss of ecosystem structure.

Pannus changes involve synovial membrane thickening and local inflammation. Studies in axenic mice have demonstrated a link between dysbiosis and arthritis. These findings have indicated that enterobacteria such as *E. coli* and Bacteroides have protective effects, but Gram-positive bacilli such as *Bifidobacterium* and *Lactobacillus* have aggravating effects [16]. Arthritis does not develop in IL-1-RA-deficient mice raised in a sterile environment; however, after recolonization with *Lactobacillus*, these mice develop severe Th17- and TLR4-dependent arthritis. Diverse data indicate that changes in microbiota composition are involved in—and may be sufficient to trigger—RA [16].

The involvement of dysbiosis, characterized by decreased microbiota diversity and the disappearance of certain families of commensal bacteria, in inflammatory pathologies is increasingly being recognized. GALT and commensal bacteria cross-talk continuously. GALT development depends on colonization by gut bacteria during the first days of life [11]. Two hypotheses have been proposed regarding the role of dysbiosis in inflammatory joint pathologies:(a)Dysbiosis might cause an excess of CD4 T cells and Th17 lymphocytes that migrate to the joints.(b)Dysbiosis might promote the migration of certain bacteria through the digestive mucosa and the translocation of these bacteria to the joint foci, as occurs in reactive arthritis.

Even though axenic mice have been studied to assess the role of the microbiota and how the immune system is influenced by the microbiome, we must keep in mind that mice are not human; arthritis models do not fully reproduce human RA or other types, microbial species and immune systems differ across species (the latter being underdeveloped in this type of mouse) and, among other factors, we have to consider artificial colonization, the influence of genetics, lifestyle, and infections among humans. Therefore, results from studies with axenic mice must be treated with caution [17].

## 4. Rheumatoid Arthritis

RA is the prototype of rheumatic diseases, characterized by inflammation and progressive joint damage. The prevalence worldwide is around 0.5–1% and women are predominantly affected (2–3 times more than a men) [18,19]. The clinician must be aware of this clinical diagnosis when joint swelling, joint tenderness can be observed in association with the presence of autoantibodies such as rheumatoid factor (RF) or anti-citrullinated protein antibody (ACPA). Structural joint damage could be visualized on conventional radiographs or other imaging techniques overtime. For the diagnosis of the disease the ACR/EULAR has stablished the classification criteria to identify patients with unexplained inflammatory arthritis of peripheral joints and no other causes can be suspected, including those influenced by age (osteoarthritis), metabolic disorders (gout) or even infectious arthritis (Lyme disease) [20]. In the pathogenesis, RA susceptibility is genetically determined and the human leucocyte antigen (HLA) class II region encoding the HLA-DRβ1 (shared epitope) molecules is the strongest genetic risk factor. Other non-HLA loci are associated with genetic risk susceptibility; of these, PTPN22 and CTLA4 are two of the most important. Among environmental influences cigarrette smoking plays a synergistic role with the genetics. Inflammation and stress at mucosal surfaces induced by environmental exposures can contribute with the disease initiation in susceptibility people [18].

A high frequency of gut microbiota alterations has been found in patients with RA [21], and individuals with dysbiosis manifest a more active form of the disease. Through DNA sequencing techniques, the fecal DNA of 44 patients with early-stage RA and healthy individuals was compared. Those with early RA had a predominance of *Prevotella copri* in their gut microbiota, whereas Bacterioides species were present in much lower abundance than observed in the healthy participants [22]. Other studies in experimental animal models of arthritis have highlighted that the transplantation of a microbiota rich in *Prevotella copri* induces RA in mice and a significant T17 lymphocyte response [23]. A Th1 cell-mediated immune response against *P. copri* has been observed in 42% of people with early-stage RA. In comparison, this response is exceptional in healthy individuals, it specifically only affects RA and it is not found in other inflammatory diseases without changes in other bacteria that make up the digestive microbiota [23]. Two subgroups were identified:(a)The first group had a Th17 and IgA response directed against the Pc-p127 peptide of *P. copri* or against the complete microorganism, showing the presence of ACPA.(b)The other group had a Th1 response and IgG antibodies against *P. copri*.

These results suggest that RA might be promoted by a microbiota rich in *P. copri*. Further study of the prevalence of *P. copri* in people with RA will be essential to determine its possible causal role in disease development. A hypothesis of molecular mimicry between these organisms and autoantigens has been proposed [24]. Differing hypotheses have been suggested regarding dysbiosis involvement in RA. The combination of genetic and environmental factors, particularly the predominance of *P. copri*, might lead to dysbiosis and subsequent mucosal alterations and local inflammation, thus stimulating Th17 to the detriment of Tregs. The resulting tolerance loss might lead to an adaptive immune response that causes RA [24,25,26]. Figure 3 illustrates how dysbiosis, particularly the over-representation of *Prevotella copri* and the reduction in Bacteroides species, may alter the intestinal mucosa and promote inflammatory responses associated with RA. Enterocytes under normal and inflamed conditions highlight changes in barrier function. Also, M cells and dendritic cells interact with microbial antigens in the mucosa, leading to activation of T helper cells in the lamina propria. Notably, Th17 cells are shown migrating through the submucosa and entering the circulation, contributing to systemic inflammation. These cells may later return to the gut as CD4 memory T cells, sustaining the inflammatory loop. This mechanism reflects how changes in the gut microbiota may promote systemic autoimmunity through disruption of mucosal homeostasis and Th17-driven responses.

Taking the genetic susceptibility to RA, several works have suggested the pathogenic role of those HLA-DRB1 alleles in mucosal dysbiosis risk. Genes coding MHC class II expressed on epithelial cells can also modulate gut microbiota composition. However, the genetically driven mechanisms of dysbiosis remain unclear [27,28].

### 4.1. Interrelationship Between CP and RA

Since the beginning of the 20th century, studies have shown improved RA symptoms during treatment for periodontal infections. It is currently accepted that oral health, specifically the oral microbiota, plays a role in RA progression. A recent study examining the composition of oral and fecal microbiota samples from patients with RA suggested that both microbiotas present dysbiosis that partially resolves after RA treatment [23]. The oral–intestinal axis is therefore a key factor responsible for the onset of this joint disease. Individuals with CP have a doubled risk of developing RA. In contrast, individuals with RA are twice as likely to become edentulous [29]. In a 2014 study, periodontal treatment for a patient with early RA and CP resulted in complete and lasting recovery [30]. Therefore, in some early cases of RA, prompt treatment for periodontal infection may induce disease regression and thus prevent the development of chronic and progressive RA. The most convincing hypothesis regarding the role of CP is that certain bacterial species present in the oral cavity might induce tolerance breakdown by exerting abnormal citrullination activity [23]. Citrullination, the post-translational modification of proteins, and both the oral and intestinal microbiota are implicated in various pathologies, including autoimmune and inflammatory diseases. Imbalances in microbiota—or dysbiosis—can lead to abnormal immune responses and chronic inflammation, while citrullination, catalyzed by enzymes present in microbiota, can generate modified antigens that may trigger autoimmune reactions. RA is strongly associated with the presence of ACPA, and the oral microbiota, particularly certain bacteria associated with periodontitis, is implicated in the citrullination process and disease onset. Studies suggest that intestinal dysbiosis, with an increase in certain bacteria such as *Prevotella copri*, may also play a role. Some bacteria, particularly those found in the mouth, express enzymes called peptidylarginine deiminases (PADs), which catalyze protein citrullination. Citrullination and microbiota, both oral and intestinal, are important factors in the pathogenesis of various diseases. A better understanding of the interactions between these elements opens promising avenues for the development of new diagnostic and therapeutic strategies [31,32].

RA autoimmunity is therefore triggered or reinforced by specific oral bacteria that cause periodontal disease. The Gram-negative bacterium *Porphyromonas gingivalis* is the main candidate in the association between CP and RA [33]. Figure 4 shows how the oral microbiota may contribute to the development of RA. The image highlights the role of specific bacteria commonly found in chronic periodontitis—such as *Porphyromonas gingivalis*, *Anaeroglobus geminatus*, and *Prevotella intermedia*—in the abnormal process of citrullination. Through enzymes like PADs, bacteria can modify host proteins, leading to the formation of citrullinated peptides such as citrullinated cytokeratin 13 (cCK13). The production of autoantibodies like ACPA and cCK13-specific antibodies is followed, which are associated with joint inflammation and disease progression in RA. Also, innate immune cells such as monocytes, neutrophils, eosinophils, and basophils may be activated by these immune complexes and contribute to systemic inflammation. Altogether, the diagram reflects the potential link between oral dysbiosis, immune activation, and the progression of RA.

RA and ACPA levels are correlated with antibody response to *P. gingivalis,* a key bacterium in the dysbiotic microbiota. However, this bacterium is not the only species implicated in this disease. The principle of microbial polysynergy must also be considered. For example, the serological response against *Prevotella* intermedia is associated with a novel ACPA directed against cCK13, a citrullinated cytokeratin peptide, detected in gingival fluid and showing high antibody responses in patients with RA. Unlike other ACPA, this autoantibody is not correlated with a serological response against *P. gingivalis*; consequently, ACPA with different specificities might be associated with different periodontal pathogens [34]. To generate these specific ACPA, *P. intermedia* might, for example, facilitate human peripheral artery disease activity in the periodontium by inducing local inflammation or acting on activated neutrophils, which are sources of citrullinated antigens. However, the lack of association between anti-cCK13-1 antibodies and CP suggests that gingival fluid is not the sole source of this antigen. In fact, cCK-13 is also expressed in the lungs and intestine. The citrullination of this peptide in the mucosal epithelium core of these organs might also be associated with dysbiosis, contributing to the generation of autoantibodies in RA [34].

#### *P. gingivalis* and Its Relationship with RA

*P. gingivalis* is present in virtually all individuals, and its proportion is directly associated with periodontal disease severity, regardless of RA status. In one study, 72% of participants in the healthy control group were positive for anti-*P. gingivalis,* compared with 63.3% with recent-onset RA and 50% with chronic RA. These findings might reflect the effectiveness of the immune system in healthy people and the inability of some people with RA to mount an immune response against *P. gingivalis*. However, relevant biological tests lack sensitivity and specificity. The prevailing hypothesis is that *P. gingivalis* might serve as an environmental trigger for RA in susceptible individuals [35]. Furthermore, two Gram-negative bacteria have been found in patients with recent-onset RA, *P. intermedia* and *Leptotrichia* spp., which have a proinflammatory ability. Moreover, a strictly anaerobic Gram-negative bacterium, *Anaeroglobus geminatus*, has been found to significantly correlate with serum RA and ACPA titers; therefore, these three bacteria might have potential roles in RA onset [36]. Furthermore, a comparison of fecal, dental, and saliva samples from patients with RA and healthy individuals revealed oral and intestinal dysbiosis in RA. In terms of Gram-negative species, the microbiota of patients with RA was found to be enriched in *Prevotella* and *Leptotrichia*. However, *Haemophilus* species were underrepresented in the oral and intestinal compartments. In terms of Gram-positive species, the study indicated a higher abundance of *Lactobacillus salivarus* in the microbiota in RA patients [37]. Moreover, the Gram-positive bacterium *Cryptobacterium curtum* has been identified as the predominant species in the microbiome of these patients [38]. This observation is interesting because *C. curtum* can citrullinate free arginine. However, its role might be controversial because ACPAs target citrullinated proteins and not free citrulline [33].

A comparison of the microbiota between patients with RA and those with osteoarthritis, aiming to determine specific correlations with the autoimmune aspect of the disease, demonstrated similar microbiota profiles between both conditions. However, no correlation was observed between serum ACPA levels and the abundance of RA-associated bacteria, such as *P. gingivalis*, *A. germinatus, Haemophilus*, or *Aggregatibacter*. Bacterially induced ACPA production might be greater early on in the disease before progressively lessening as autoantibody production in the joints is established. Concordantly, an underrepresentation of *Porphyromonas*, *Prevotella*, and *Treponema* species has been observed in patients with RA and CP [37,38].

### 4.2. Relationship Between RA and Infection

The roles of infectious agents in RA development have been studied since the 19th century; while definitive findings remain lacking, several hypotheses have been proposed. Bacteria in the digestive tract might have direct roles in citrullination, a phenomenon involved in the catalysis of citrulline from the transformation of arginine by PAD enzymes. Patients with RA frequently experience periodontitis, often in a severe form associated with *P. gingivalis*. Furthermore, citrullinated proteins have been found in the periodontium of patients with periodontitis; a relationship has been demonstrated between a high presence of ACPA antibodies and periodontitis in patients with RA, as previously discussed. These studies have confirmed the presence of *P. gingivalis* in large quantities in the oral microbiota and *Prevotella* in the digestive microbiota of patients with RA. These two families express PADs [39]. Another hypothesis is based on the association between intestinal bacterial microbiota and tolerance breakdown. Studies in axenic animals with immature immune systems include experiments with mice containing only proinflammatory filamentous bacteria. These mice develop an excess of CD4 and Th17 T lymphocytes and are prone to arthritis, whereas other mice with an intestinal microbiota composed only of *Bacteroides fragilis* show overrepresented regulatory T lymphocytes or Tregs and arthritis resistance. These experiments have demonstrated that dysbiosis promotes the onset of polyarthritis [39]. In addition, the detection of DNA from different microorganisms, particularly *P. gingivalis*, in the synovial tissue and circulating blood in patients with polyarthritis suggests that the presence of bacterial debris or microorganisms in the joints might be associated with RA onset. These hypotheses are not mutually exclusive and might have roles in the preclinical or clinical phases. Bacteria in the oral and digestive microbiota might play a role in the preclinical phase. This is particularly true when ACPA antibodies appear, which may occur several years before disease onset, whereas the translocation of bacteria to the joints might play a role in the clinical phase. Confirming these hypotheses might enable the development of new therapeutic approaches such as microbiota modulation through the administration of probiotics, bacterial control in the oral cavity, or antibiotics [33,40].

### 4.3. RA and Intestinal Wall Weakness

In recent years, studies have indicated strong links between abnormalities in the intestinal microbiota and RA. Increases in the populations of certain types of harmful bacteria have often been associated with disease severity. However, the exact mechanism through which the gut bacteria influence joint inflammation remains unclear. Several mechanisms have been considered, such as the gut bacteria modulating the development of specific inflammatory cells responsible for arthritis or particular bacterial metabolites contributing to disease severity. An alternative causal hypothesis focuses on the links between arthritis severity and bacterially induced gut wall weakening. Mice with a genetic predisposition to leaky gut have been shown to develop signs of severe arthritis. Another mouse model engineered to develop collagen-induced arthritis showed diminished joint inflammation after gut permeability improvement [41,42,43,44,45,46,47]. Figure 5 illustrates how dysbiosis, both oral and intestinal, can contribute to joint inflammation by promoting intestinal permeability. Microbial metabolism leads to the production of inflammatory molecules such as LPS, lipopolysaccharide-binding protein (LBP), and intestinal fatty acid-binding protein (IFABP), which are associated with disruption of the epithelial barrier. Dendritic cells and macrophages respond to these molecules, activating the immune and local inflammatory responses in the gut. The condition known as “leaky gut” allows microbial components to cross into the bloodstream, where they may contribute to systemic inflammation and increased RA severity. Antimicrobial peptides, bacteria-specific IgA, and activated immune cells are also represented as part of the intestinal defense mechanisms. Overall, the image underscores how alterations in the microbiota and their byproducts can affect gut integrity and play a role in the worsening of autoimmune joint disease.

There are numerous microbial metabolites implicated in the inflammatory processes of RA, one of which is the SCFAs; butyrate; propionate; and acetate, which are produced through the fermentation of dietary fiber by anaerobic bacteria. SCFAs promote intestinal homeostasis by reinforcing epithelial tight junctions (e.g., via occludin and ZO-1) and modulate the differentiation of regulatory T and B cells, sup-pressing NF-κB activity and reducing the production of proinflammatory cytokines such as Fas-α and IL-1β. In arthritis models, butyrate has been shown to reduce inflammation, by acting as an endogenous histone deacetylase (HDAC) inhibitor, and improve bone density. Additionally, SCFAs help maintain intestinal barrier integrity by decreasing intestinal permeability and the translocation of proinflammatory molecules, thereby preventing systemic immune activation [7,48]. Additionally, these microbiome-derived metabolites have been shown to activate the mammalian target of the rapamycin (mTOR) pathway, which is important for regulating protein synthesis and muscle hypertrophy [48].

On the other hand, LPS, a component of Gram-negative bacteria membranes, acts as a potent proinflammatory molecule. When gut permeability increases (often due to dysbiosis), LPS can cross the bloodstream and activate the TLR4 signaling pathway. Elevated circulating levels of LPS and its LBP have been observed in RA patients and correlate with disease severity. Disruption of gut barrier integrity promotes LPS leakage into systemic circulation, enhancing macrophage activation, Th17 cell differentiation, and cytokine release (e.g., IL-6, IL-17), in turn contributing to synovial inflammation and joint destruction [6].

People with RA have elevated blood levels of LPS, LBP, and the intestinal fatty acid-binding protein. All these molecules are known biomarkers of intestinal damage, and lipopolysaccharide-binding protein levels have been closely correlated with disease severity [49]. In RA, the intestinal mucosa is profoundly damaged and lacks barrier function, and a build-up of white blood cells in the gut causes inflammation. When bacteria cross the boundary of the intestinal mucosa, repairing intestinal permeability defects with specific drugs has been found to inhibit joint inflammation. Current research findings have not yet fully explained the chain of mechanisms linking this intestinal wall weakening to arthritis [50,51]. Therefore, although modulation of the degree of intestinal permeability has been directly associated with joint inflammation, missing links in this relationship must be elucidated to enable the gut to be a useful therapeutic target [52,53,54,55]. Notably, ameliorating intestinal permeability has been suggested as a new model for effective treatment. In contrast, the use of probiotics/prebiotics to restore intestinal barrier integrity might prevent intestinal permeability or inhibit the movement of inflammatory cells into and out of the intestine, thereby decreasing arthritis severity [56,57,58,59].

Unfortunately, although modulating the degree of intestinal permeability has been demonstrated to be directly associated with joint inflammation, missing links in this relationship must be determined. Although the specific dysbiotic microbiota associated with RA remains to be clarified, in the absence of basic treatment, near-systematic disruption of the oral and intestinal microbiota is observed during this disease. Therefore, dysbiosis is considered to be a key factor in perpetuating inflammation and breaking down citrullinated protein tolerance [60,61,62]. Interestingly, although underrepresented in the oral microbiota in some studies, *P. gingivalis* appears to provoke inflammatory responses by creating an environment conducive to the persistence of dysbiotic bacteria; however, it also enables the accumulation of citrullinated proteins in gingival tissues. Therefore, *P. gingivalis* is a key bacterium in RA pathogenesis. Research findings have suggested that the gut and oral cavity might be a useful therapeutic target. In particular, ameliorating intestinal permeability might constitute a new model for effective treatment with drugs that restore intestinal barrier integrity and decrease disease severity in preclinical models [63,64].

An altered gut microbiota can serve as a diagnostic and predictive biomarker for inflammatory arthritis, including RA. Several studies have shown that the composition and function of the gut microbiota are altered in patients with inflammatory arthritis, and these changes could be used to identify at-risk individuals and predict disease progression. Although further research is needed to validate its role as a diagnostic and predictive biomarker, current data suggest that altered gut microbiota plays an important role in the development and progression of inflammatory arthritis. Identifying specific microbial signatures could lead to earlier diagnosis and more personalized treatment for this disease [48,65,66,67,68].

Unfortunately, there are currently no recommendations for treating RA based on findings related to intestinal microbiota or its influence on inflammatory processes in joints. Some nutritional studies on the use of probiotics as adjuvants and their effect on the microbiota in RA have been previously published. These studies demonstrate the favorable impact of increased SCFAs derived from microbial fermentation on the maintenance of the epithelial barrier, however, they are not included in the therapeutic guidelines for the disease [69]. According to Badsha, supplementing with *Lactobaciullus casei* alongside baseline treatment led to a reduction in inflammatory cytokines and C-reactive protein in RA [70].

## 5. Ankylosing Spondylitis, Associated Pathologies and Microbiota

Several pathologies are grouped under the term spondylorthritis (SPA), such as ankylosing spondylitis and psoriatic arthritis. Psoriasis may appear simultaneously with or precede joint pain. Approximately 15–20% of people with SPA have cutaneous involvement pathologies of the osteoarticular system associated with cutaneous manifestations. There is also reactive arthritis, which is recognized by three symptoms: urethritis, conjunctivitis, and arthritis manifestations, as well as enterocolopathy, with gastrointestinal manifestations that can range from intestinal permeability disorder without major symptoms to Crohn’s disease or hemorrhagic enterocolitis [71,72].

The gut microbiota hypothesis is increasingly considered to explain SPA pathogenesis [73]. Among all these pathologies, one common thread is the association with HLA-B27, especially in RA, where the prevalence of affected individuals with this allele exceeds 90%. It is also used to aid diagnosis. It is hypothesized that a microbial infection could be involved in the onset or maintenance of SPA in people with a genetic predisposition to the condition. This infection of the mucosal membranes (e.g., the intestine) would initially cause reactive arthritis, which in many cases would lead to SPA. This hypothesis is consistent with a second hypothesis suggesting the role of the oral cavity and the intestine, more specifically its inflammation, which is considered a pathophysiological component of SPA [72,73].

Some clinical manifestations are common to all SPA. Enthesis—more specifically fibrocartilaginous enthesis—is the main target of this group of diseases. We can cite an enthesis known as the Achilles heel, which inserts into the calcaneus bone. The enthesis is described as the interface between ligaments or tendons and bone. Two types of entheses have been defined: fibrous and fibrocartilaginous [72,74]. Enthetic regions are richly vascularized and innervated by various receptors, such as those for pain or deep sensitivity (proprioceptive receptors). It is also important to highlight that these areas present high metabolic activity. The hypotheses discussed in several studies [72,74] tend to explain that mechanical stress on these entheses could cause microtrauma and inflammation, leading to the expression of genes encoding inflammatory cytokines. This inflammation, in turn, could be a factor involved in bone formation and, therefore, in the calcification of these pathologies. In fact, certain cellular and molecular inflammatory factors, such as macrophages and TNFα, are believed to promote bone formation. These changes, called enthesitis ossificans, may be partly responsible for symptoms such as stiffness, vertebral ankylosis, sacroiliitis, or vertebral syndesmophytes [72,73,74].

In ankylosing spondylitis, gastrointestinal manifestations have been shown to precede or occur after diagnosis. Therefore, it is very important to study the role of the oral and intestinal microbiota. Studies of RA versus the oral and intestinal microbiota have shown the presence of inflammatory histological abnormalities in 60% of patients, dysbiosis during this disease, and the presence of inflammatory biological markers such as LPS present in Gram-negative bacteria, IL-23, and S100 proteins. This indicates immune system and microbiota activation [72,73,74].

## 6. Conclusions

The immune system is regulated by the interconnected responses of various factors, where eubiosis or dysbiosis affects not only defense against pathogens but also inflammatory processes. More specifically, the gut microbiota maintains a bidirectional relationship with the oral microbiota, which contributes to immune response abnormalities. In this regard, it has been shown that an increase in *P. copri* and a decrease in Bacteroides contribute to proinflammatory pathways in RA patients. *P. copri* primarily affects dysbiosis, increasing intestinal permeability and stimulating Th17 release. This causes local and, subsequently, systemic inflammation. Therefore, a molecular mimicry is described between the pathogens responsible for dysbiosis and the host’s autoantigenic response. Finally, dysbiosis and its impact on inflammatory pathways and immune responses in RA contribute to disease severity. In this context, probiotics and dietary changes have been investigated as possible treatment approaches for RA. The many benefits of probiotics include its possibility of reversing gut inflammation, its prevention of side effects associated with non-steroidal anti-inflammatory drugs, and its ability to help in modulating the negative effects of dysbiosis.

## Figures and Tables

**Figure 1 healthcare-13-01942-f001:**
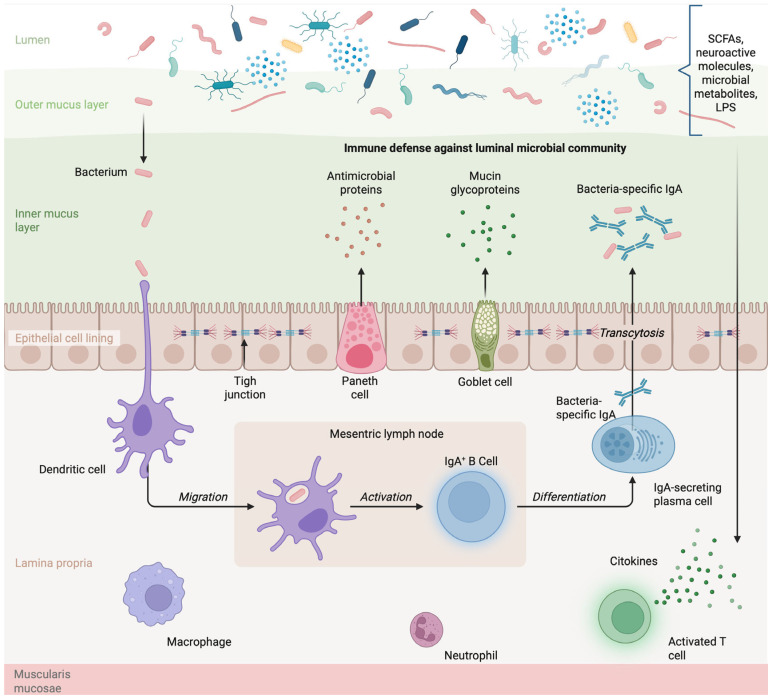
The gut microbiome and immune system. The different tissues that make up the intestinal barrier are shown, including specialized cells such as Paneth, goblet, and immune system cells, which respond to microbial factors and activate the immune system. Created in BioRender. Delgado, D. (2025) https://BioRender.com/prcdink, accessed on 6 June 2025.

**Figure 2 healthcare-13-01942-f002:**
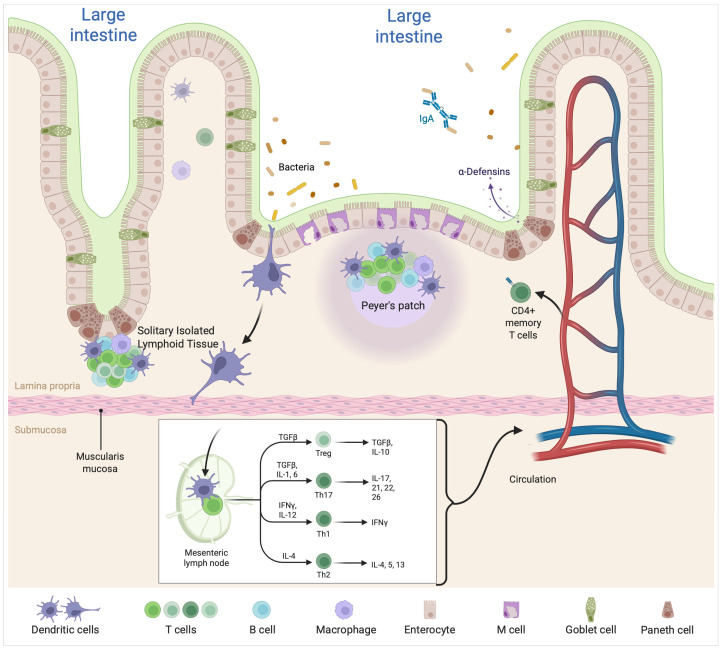
Gut-associated lymphoid tissue (GALT). The GALT is covered by M cells, as is the Peyer’s patch in the small intestine. An increased presence of induced lymphoid tissue (SILT) is observed in the colon. Created in BioRender. Delgado, D. (2025) https://BioRender.com/xpu2nq4, accessed on 6 June 2025.

**Figure 3 healthcare-13-01942-f003:**
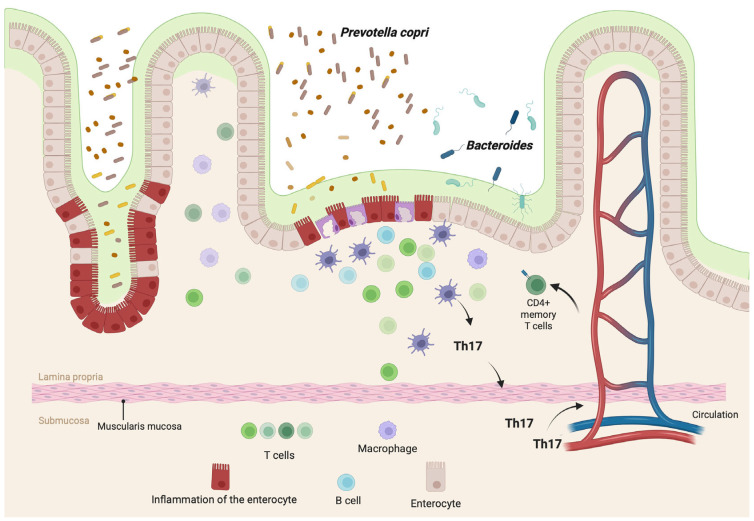
Dysbiosis and RA. Changes in the intestinal microbiota, such as an increase in *P. copri* and a deterioration in Bacteroides, trigger changes in the intestinal mucosa due to increased inflammation. Th17, T helper 17. Created in BioRender. Torres, D. (2025) https://BioRender.com/dzpnjhl, accessed on 6 June 2025.

**Figure 4 healthcare-13-01942-f004:**
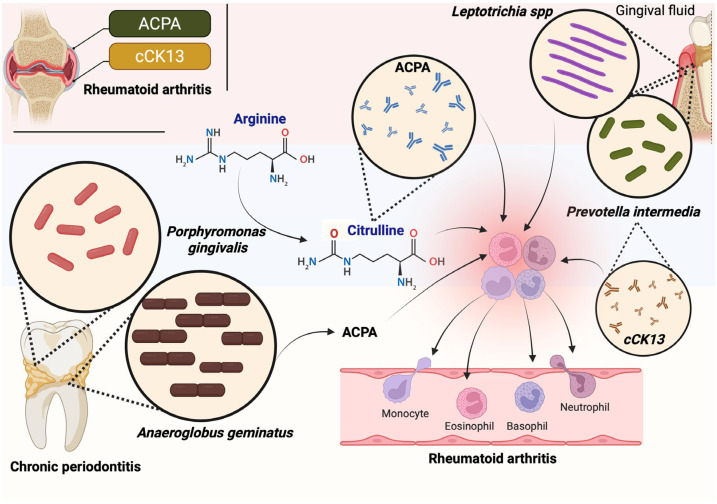
Oral microbiota and RA. Oral dysbiosis is associated with increased citrulline, which is involved in the release of ACPA and cCK13 antibodies, which are associated with the progression of AR due to increased systemic inflammation. ACPA, anti-cyclic citrullinated peptide; cCK13, anti-citrullinated cytokeratin 13. Created in BioRender. Torres, D. (2025) https://BioRender.com/zgsds4t, accessed on 20 June 2025.

**Figure 5 healthcare-13-01942-f005:**
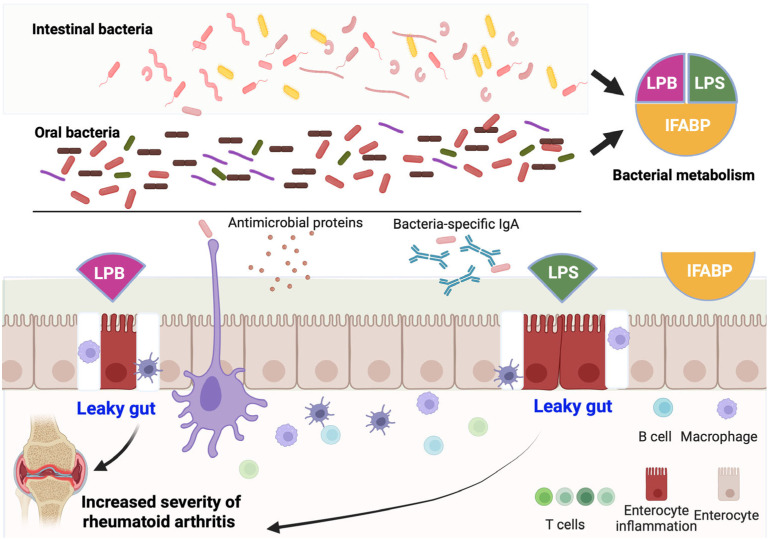
Dysbiosis and leaky gut. Intestinal and oral dysbiosis, by itself or via derived metabolites, increases the intestinal inflammatory response, favoring intestinal permeability, which has implications for increased severity in RA patients. Created in BioRender. Torres, D. (2025) https://BioRender.com/ub0mbis, accessed on 6 June 2025.

## Data Availability

No new data were created or analyzed in this study; thus, data sharing is not applicable.

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
