# Peer review of "Inflammatory Joint Pathologies and the Oral–Gut Microbiota: A Reason for Origin"

_healthcare, 2025, doi:10.3390/healthcare13161942_

Round 1
Reviewer 1 Report
Comments and Suggestions for Authors
The authors have carried out a narrative review associating oral and gut microbiota with inflammatory arthritis. The following points could be considered by authors for improvement:
- The authors have alluded to only rheumatoid arthritis but psoriatic arthritis and ankylosing spondylitis are other inflammatory joint pathologies that need to be mentioned and discussed, particularly on their associations with oral-gut microbiota.
- The narrative should include detailed assessment of the role of bacterial metabolites such as short chain fatty acids and lipopolysaccharide with joint inflammation.
- There should also be discussions around genetic susceptabilities, particularly with HLA and NOD polymorphisms and their association with microbiota with inflammatory arthritis.
- Can altered microbiota be used as diagnostic/predictive biomarker for inflammatory arthritis? Debate with appropriate citations.
- Critically appraise the articles included in this narrative review.
- Use SANRA, the Scale for the Assessment of Narrative Review Articles, for assessing the quality of this narrative review and provide a checklist related to this study.
Author Response
|
Reviewer #1 comments |
|||||
|
Comment number |
Page number |
Line number |
Comment |
ANSWERS |
|
|
1 |
13 |
448 to 488 |
The authors have alluded to only rheumatoid arthritis but psoriatic arthritis and ankylosing spondylitis are other inflammatory joint pathologies that need to be mentioned and discussed, particularly on their associations with oral-gut microbiota.
|
The information was included as indicated by the reviewer.
|
|
|
2 |
12 |
386 to 407 |
The narrative should include detailed assessment of the role of bacterial metabolites such as short chain fatty acids and lipopolysaccharide with joint inflammation |
The information was included as indicated by the reviewer.
|
|
|
3 |
8 |
244 to 250 |
There should also be discussions around genetic susceptabilities, particularly with HLA and NOD polymorphisms and their association with microbiota with inflammatory arthritis |
The information was included as indicated by the reviewer.
|
|
|
4 |
13 |
439 to 447 |
Can altered microbiota be used as diagnostic/predictive biomarker for inflammatory arthritis? Debate with appropriate citations |
The information was included as indicated by the reviewer.
|
|
|
5 |
- |
- |
Critically appraise the articles included in this narrative review |
The necessary changes were made to improve this point
|
|
|
6 |
- |
- |
Use SANRA, the Scale for the Assessment of Narrative Review Articles, for assessing the quality of this narrative review and provide a checklist related to this study |
Thank you for guiding us through the use of this tool. We performed the exercise and received a score of 10 out of 12 points on the scale. |
|
|
The article includes additional changes that are not in line with your comments. These changes were suggested by other reviewers. |
|
||||
Reviewer 2 Report
Comments and Suggestions for Authors
The manuscript presents interesting content, but several critical issues need to be addressed to improve its scientific value and clarity.
Firstly, the review lacks a clear methodological structure. No information is provided regarding the inclusion/exclusion criteria, time frame, or source selection strategy, which limits the reproducibility and reliability of the work. Moreover, the narrative is often excessively descriptive rather than analytical, and the manuscript fails to offer a critical comparison of studies or highlight existing knowledge gaps and conflicting data.
Many of the pathophysiological assumptions are based on preclinical studies in axenic mice, yet these are often generalised without caution to human conditions. This overinterpretation should be balanced by a more rigorous evaluation of translational relevance.
Additionally, key concepts—such as the role of P. gingivalis, the mechanisms of citrullination, and the "oral–gut–joint" axis—are repeated without deepening the clinical implications or proposing an integrated pathogenic model. The figures, while visually clear, are not conceptually original and do not help the reader navigate the complexity of the topic.
Finally, the conclusion lacks specificity and fails to provide clear directions for clinical practice or research priorities.
We strongly suggest citing the following relevant and recent article, which addresses the emerging role of the microbiome in musculoskeletal recovery and offers a translational perspective:Microbiome in motion: Revolutionizing musculoskeletal recovery (DOI: 10.1097/ph9.0000000000000057).
Author Response
|
Reviewer #2 comments |
|
||||
|
Comment number |
Page number |
Line number |
Comment |
Answer |
|
|
1 |
3 |
123 to 131 |
Firstly, the review lacks a clear methodological structure. No information is provided regarding the inclusion/exclusion criteria, time frame, or source selection strategy, which limits the reproducibility and reliability of the work. Moreover, the narrative is often excessively descriptive rather than analytical, and the manuscript fails to offer a critical comparison of studies or highlight existing knowledge gaps and conflicting data |
The search criteria used for this review are described. Additionally, a review of studies related to the objective of this work was included.
|
|
|
2 |
6 |
206 to 212 |
Many of the pathophysiological assumptions are based on preclinical studies in axenic mice, yet these are often generalised without caution to human conditions. This overinterpretation should be balanced by a more rigorous evaluation of translational relevance |
The information was included as indicated by the reviewer.
|
|
|
3 |
8 to 9 |
265 to 279 |
Additionally, key concepts—such as the role of P. gingivalis, the mechanisms of citrullination, and the "oral–gut–joint" axis—are repeated without deepening the clinical implications or proposing an integrated pathogenic model. The figures, while visually clear, are not conceptually original and do not help the reader navigate the complexity of the topic. |
The work performed in this section aligns with the reviewer's recommendations. Unnecessary figures were also eliminated.
|
|
|
4 |
14 |
498 to 512 |
Finally, the conclusion lacks specificity and fails to provide clear directions for clinical practice or research priorities
|
The work performed in this section aligns with the reviewer's recommendations.
|
|
|
5 |
12 |
386 to 399 |
We strongly suggest citing the following relevant and recent article, which addresses the emerging role of the microbiome in musculoskeletal recovery and offers a translational perspective:Microbiome in motion: Revolutionizing musculoskeletal recovery (DOI: 10.1097/ph9.0000000000000057)
|
We are grateful for the literary contribution, which the reviewer indicated should be included. We believe it is important to strengthen this work. |
|
|
The article includes additional changes that are not in line with your comments. These changes were suggested by other reviewers. |
|||||
Reviewer 3 Report
Comments and Suggestions for Authors
In healthcare-3716564, Salazar-Páramo et al discuss the relationship between the inflammatory joint pathologies and the oral-gut microbiota. The topic of this review is interesting and fits well the scope of Healthcare. The reviewer feels it can be accepted after some minor amendments.
(1) Line 41: Set the number 14 as a superscript.
(2) The authors might have used AI to prepare this review. The use of Arabic numeral ordering should be avoided in the main body of the text. Lines 74-80; 147-150
(3) The authors need to add a new paragraph and discuss the future perspectives: how to develop a strategy to manage rheumatoid arthritis through modulating the oral-gut microbiota.
Author Response
|
Reviewer #3 comments |
|
||||
|
Comment number |
Page number |
Line number |
Comment |
Answer |
|
|
1 |
1 |
41 |
Line 41: Set the number 14 as a superscript. |
The exponential number 14 was corrected as indicated by the reviewer.
|
|
|
2 |
- |
74-80 and 147-150 |
The authors might have used AI to prepare this review. The use of Arabic numeral ordering should be avoided in the main body of the text. Lines 74-80; 147-150 |
The Arabic numerals indicated by the reviewer were removed throughout the article.
|
|
|
3 |
14 |
507 to 512 |
The authors need to add a new paragraph and discuss the future perspectives: how to develop a strategy to manage rheumatoid arthritis through modulating the oral-gut microbiota. |
The information was included as indicated by the reviewer. |
|
|
The article includes additional changes that are not in line with your comments. These changes were suggested by other reviewers. |
|||||
Reviewer 4 Report
Comments and Suggestions for Authors
The article addresses a topic of relevance to the biomedical field: the relationship between microbiota and inflammatory joint diseases, especially rheumatoid arthritis; however, it presents significant structural and methodological deficiencies that must be addressed before considering publication. It is recommended to carry out an in-depth review of the abstract, introduction and general writing, as well as to include a clear methodological section and a more precise delimitation of the purpose of the review. The improvement in these aspects will bring greater rigor and academic value to the manuscript.
Comments:
- It is recommended that the central problem that motivates the review be defined more clearly.
- Including a question or purpose of review would better guide the reader.
- The introduction should provide a more robust critical justification.
- The procedure followed for the search and analysis of sources is not indicated.
- It is recommended to describe the selection criteria and justify the choice of literature included.
A file with highlighting is attached where there is no clarity or defined association with the developed topic

- A grammar and style check is recommended.
- Improving textual cohesion and avoiding redundancies will contribute to a better understanding of the text.
- It is suggested to unify the academic tone throughout the manuscript.
Author Response
|
Reviewer #4 comments |
|
||||
|
Comment number |
Page number |
Line number |
Comment |
Answer |
|
|
1 |
3 |
116 to 122 |
|
This point was improved in the work as indicated by the reviewer.
|
|
|
2 |
3 |
116 to 122 |
|
This point was improved in the work as indicated by the reviewer.
|
|
|
3 |
3 |
104 to 115 |
|
This point was improved in the work as indicated by the reviewer.
|
|
|
4 |
3 |
123 to 131 |
|
Description included as requested by reviewer
|
|
|
5 |
3 |
123 to 131 |
|
A description of the search criteria for this review was included. |
|
|
The article includes additional changes that are not in line with your comments. These changes were suggested by other reviewers. |
|||||
Round 2
Reviewer 1 Report
Comments and Suggestions for Authors
Thanks for the revision
Author Response
Comments 1: Thanks for the revision
We appreciate your valuable contributions to this work.
Reviewer 2 Report
Comments and Suggestions for Authors
thank you
Author Response
Comments 1: thank you
Response 1: We appreciate your valuable contributions to this work.
Reviewer 4 Report
Comments and Suggestions for Authors
Dear Editor,
Here you have my evaluation of manuscript healthcare-3716564 .
Final evaluation of the article:
This article on the relationship between microbiota and inflammatory joint diseases, with a particular focus on rheumatoid arthritis, has been substantially improved in terms of clarity and structure after the corrections suggested in the initial review were made. The requested modifications to the abstract, introduction, and general wording have been adequately implemented, providing a better theoretical framework and making the information more accessible and coherent.
Additionally, the newly presented methodological section has effectively clarified the focus of the review. This improvement contributes to the transparency and replicability of the study and the approaches discussed. The purpose of the review is now more precisely defined, providing readers with clearer guidance on the article's specific objectives.
The article has improved significantly, but a final revision is suggested to correct minor wording issues and ensure consistent use of technical terminology. The content now meets the publication standards of a high-impact scientific journal.
Therefore, I recommend acceptance of the article, with the aforementioned observations, and I am confident that the final version will be of great use to the biomedical community, especially in the area of microbiota and inflammatory joint diseases.
Author Response
Comments 1: The article has improved significantly, but a final revision is suggested to correct minor wording issues and ensure consistent use of technical terminology. The content now meets the publication standards of a high-impact scientific journal.
Response 1: Thank you very much for your valuable guidance. The suggested changes have been made throughout the article. We have attached the manuscript with these modifications. We have also enclosed the English language correction certificate.
